# Candle Soot as a Novel Support for Nickel Nanoparticles in the Electrocatalytic Ethanol Oxidation

**DOI:** 10.3390/nano14121042

**Published:** 2024-06-18

**Authors:** Muliani Mansor, Siti Noorleila Budiman, Azran Mohd Zainoodin, Mohd Paad Khairunnisa, Shinya Yamanaka, Nurfatehah Wahyuny Che Jusoh, Shahira Liza

**Affiliations:** 1Department of Chemical and Environmental Engineering, Malaysia-Japan International Institute of Technology (MJIIT), Universiti Teknologi Malaysia Kuala Lumpur, Jalan Sultan Yahya Petra, Kuala Lumpur 54100, Malaysia; muliani94@graduate.utm.my (M.M.); sitinoorleila@gmail.com (S.N.B.); nurfatehah@utm.my (N.W.C.J.); 2Fuel Cell Institute, Universiti Kebangsaan Malaysia, Bangi 43600, Malaysia; azrans@ukm.edu.my; 3Department of Applied Science, Muroran Institute of Technology, Muroran 050-8585, Japan; 4Tribology and Precision Machining i-Kohza, Malaysia-Japan International Institute of Technology (MJIIT), Universiti Teknologi Malaysia Kuala Lumpur, Jalan Sultan Yahya Petra, Kuala Lumpur 54100, Malaysia; shahiraliza@utm.my

**Keywords:** carbon nanoparticles, Ni catalyst, direct ethanol fuel cell

## Abstract

The enhancement of carbon-supported components is a crucial factor in augmenting the interplay between carbon-supported and metal-active components in the utilization of catalysts for direct ethanol fuel cells (DEFCs). Here, we propose a strategy for designing a catalyst by modifying candle soot (CS) and loading nickel onto ordered carbon soot. The present study aimed to investigate the effect of the Ni nanoparticles content on the electrocatalytic performance of Ni–CS, ultimately leading to the identification of a maximum composition. The presence of an excessive quantity of nickel particles leads to a decrease in the number of active sites within the material, resulting in sluggishness of the electron transfer pathway. The electrocatalyst composed of nickel and carbon support, with a nickel content of 20 wt%, has demonstrated a noteworthy current activity of 18.43 mA/cm^2^, which is three times that of the electrocatalyst with a higher nickel content of 25 wt%. For example, the 20 wt% Ni–CS electrocatalytic activity was found to be good, and it was approximately four times higher than that of 20 wt% Ni–CB (nickel–carbon black). Moreover, the chronoamperometry (CA) test demonstrated a reduction in current activity of merely 65.80% for a 20 wt% Ni–CS electrocatalyst, indicating electrochemical stability. In addition, this demonstrates the great potential of candle soot with Ni nanoparticles to be used as a catalyst in practical applications.

## 1. Introduction

Fossil fuels, comprising coal, oil, and natural gas, have been the primary source of energy for more than a century and a half, presently accounting for approximately 80% of the global energy demand. The persistence of issues such as energy supply security and environmental pollution can be attributed to swift population growth and industrialization. In recent years, fuel cells have garnered significant attention as advanced energy-conversion systems due to the pressing issues of fossil fuel scarcity and environmental pollution [1]. It is anticipated that fuel cell technologies will significantly diminish the utilization of oil and the emission of pollutants, including greenhouse gases, in comparison to traditional power generation technologies reliant on combustion. There are many distinct types of fuel cells, which are categorized based on the combination of fuel and oxidant [2,3]. Among these types, direct ethanol fuel cells (DEFCs) have gained significant attention due to the advantageous properties of ethanol as a liquid fuel. These properties include non-toxicity, high volumetric energy density (6.32 kWh/L), and ease of handling [4]. Theoretically, the energy generated by DEFCs surpasses that of the pioneering direct methanol fuel cell (DMFC). Nevertheless, the extensive commercialization of this technology has yet to be achieved, primarily owing to the high costs associated with the electrode catalysts and their limited durability [5,6]. Therefore, it is significant to develop electrocatalysts that exhibit high catalytic activity toward the ethanol oxidation reaction (EOR) while utilizing cost-effective materials in order to optimize fuel cell performance.

Nickel (Ni) is considered to be the most promising non-precious catalyst among various transition-based catalysts, exhibiting significant potential in the oxidation of alcohols in alkaline conditions. Ni exhibits promising catalytic properties owing to its surface oxidation characteristics [7]. The existence of a surface oxide layer on the nickel material can serve as a catalytic reaction active site, thereby promoting the adsorption and activation of ethanol molecules on the surface of the nickel. Askari et al. [8] reported the synthesis of composites comprising Ni/NiO particles and multi-walled carbon nanotubes (MWCNTs) for the purpose of methanol oxidation. The composites exhibited uniform embedding of Ni/NiO particles, which were found to be in the range of 20–30 nm, on MWCNTs that were in the range of 10–40 nm. The composites showed enhanced electron transfer and reaction kinetics, which resulted in a peak current density of 15.94 mA/cm^2^ at a peak potential of 0.43 V. In another study, Chemcoub et al. [9] focused on the development of an electrocatalyst that is both cost-effective and efficient, utilizing poly-pyrrole (PPy) and nickel nanoparticles (NiNPs) through galvanostatic and potentiostatic methods by investigating the effect of NiNPs content. The optimized composition of the electrocatalyst was achieved at 6 mM NiNPs content, resulting in a significant increase in anodic current density from 3.58 mA/cm^2^ to 20.1 mA/cm^2^. It is found that the presence of an excessive quantity of nickel particles results in a reduction in the active sites of the material, leading to sluggishness in the electron transfer pathway. The manipulation of nickel loading has the potential to exert an impact on the catalytic activity, selectivity, and overall performance of the catalyst. An increased concentration of nickel yields a greater number of active sites that facilitate the adsorption and activation of ethanol molecules. This phenomenon has the potential to result in elevated reaction rates and enhanced conversion of ethanol to the anticipated outputs. Sayed et al. [10] synthesized a standalone Ni sulfide nano-sheets electrode in the DEFC. The conclusive results indicate that the enhanced ethanol oxidation activity can be attributed to the function of sulfur in supplying electrons to Ni, thereby augmenting the charge transfer rates and electrical conductivity. However, it is advisable to investigate the electrochemical activity of the electrode with carbon doping. Metal catalysts used in single form tend to experience a decline in their electrochemical catalytic activity during the reaction process, owing to their tendency to agglomerate into larger particles [11].

Numerous techniques have been established in the pursuit of improved and novel electrocatalysts. One common strategy to mitigate agglomeration, reduce costs, and enhance catalyst activity is by incorporating carbon support. The utilization of carbon nanostructures and their multifunctional composites has garnered significant interest in the realm of energy applications due to their excellent electrical conductivities, facile decoration, and high specific surface area [12,13]. However, most carbon materials are difficult to synthesize and costly. Sikeyi et al. [14] utilized a chemical vapor deposition technique to fabricate palladium (Pd) supported on carbon nanofibers (CNFs), with chicken oil serving as a carbon source. In alkaline conditions, the current density for ethanol oxidation exhibited by the Pd/CNFs electrocatalyst is 1160 μA/cm^2^, greater than that of the commercial Pd/C electrocatalyst, which is 440 μA/cm^2^. This means that inexpensive materials like chicken oil as a carbon source can be used as catalyst support in EOR. Mamidi et al. [15] synthesized by depositing candle soot-derived carbon nanoparticles onto metal-organic framework-based cobalt oxide nano leaves. This composite exhibits superior electrochemical properties, including high specific capacities of 811 and 503 mAh/g at 50 and 1000 mA/g current densities and good cyclic stability at 1000 mA/g current density, delivering an excellent specific charge capacity of 490 mAh/g even after 400 cycles, making it a promising candidate for high-performance and stable Li-ion batteries. Yu et al. [16] developed a bifunctional electrocatalyst based on N and S co-doped carbon nanofibers with embedded candle soot and NiCo and NiCo_2_S_4_ nanoparticles. This electrocatalyst, NiCo/NiCo_2_S_4_@CNS-800, exhibits excellent bifunctional electrocatalytic activities for both oxygen reduction reactions (ORRs) and oxygen evolution reactions (OERs), with a half-wave potential of 0.80 V for ORRs and a potential of 1.60 V at 10 mA/cm^2^ for OERs. The synergistic effects of the embedded candle soot, decorated nanoparticles, continuous fibrous networks, and incorporated N and S dopants could prevent the aggregation of decorated nanoparticles, increase the active sites, enhance mass transfer, and promote surface adsorption.

Within this particular context, the potential suitability of candle soot as an anode material may be investigated. Candle soot has emerged as a viable candidate to replace traditional carbon materials due to its simplicity and affordability [17]. Carbon nanoparticles (CNPs) can be readily produced through the process of burning candles, which is a simple and straightforward method. Candle soot carbon is the product of the combustion of paraffin wax, which generates a fractal-like interconnected porous carbon nanostructure. The carbon resulting from the burning of paraffin wax in candles is known as candle soot carbon. This carbon exhibits a unique nanostructure consisting primarily of nanospheres with varied internal structures of concentrically wrapped and graphene-like layers of carbon [18]. These carbon nanostructures have a high surface area, a mesoporous structure, and favorable electrical conductivity [19]. Furthermore, the utilization of candle soot as a supporting material has been shown to effectively inhibit the agglomeration of metal alloy nanoparticles [20]. Kanakaraj and Sudakar [19] fabricated carbon nanoparticles by turning soot pollutants from candles into an effective universal anode for metal–ion batteries. The CS-A/Ar sample exhibited a noteworthy lithium–ion storage capacity of 1200 mAh/g when subjected to a current density of 150 mA/g. Furthermore, at a significantly higher current density of 25 A/g, the sample demonstrated a storage capacity of 230 mAh/g. The researchers discovered that candle soot exhibits a fractal chain-like structure, which facilitates the establishment of optimal conductivity necessary for electrode functionality. The aforementioned study indicates that carbon soot possesses significant potential for much broader applications. Thus, candle soot has outstanding properties for functioning as a host matrix for nanoparticles such as nickel. Limited scholarly resources exist that specifically address the electrochemical properties of candle soot carbon when used as an anode material for energy storage.

In the present study, we employed the wet impregnation method to synthesize nickel supported on candle soot (Ni–CS) with different weight loadings, which serves as an electrocatalyst for ethanol oxidation. To the best of our knowledge, this novel anode catalyst material (candle soot) has not been previously reported in the literature. The electrochemical behavior of the electrocatalyst in alkaline media was assessed through electrochemical techniques such as cyclic voltammetry (CV) and chronoamperometry (CA), both with and without the presence of ethanol. The present study offers a potential avenue for the utilization of soot waste, a common pollutant, as a valuable resource for producing carbon nanomaterials with potential applications in fuel cells.

## 2. Methodology

### 2.1. Collection of Candle Soot Nanoparticles

In a basic laboratory configuration, a candle (Jelita 1280, Selangor, Malaysia), with a weight of 42 g for each candle stick, was ignited. The direct-burning method was employed to deposit the candle soot (specifically, the tip flame soot) onto the ceramic lid [17]. The carbon yield for each batch of CS is around 3 g per hour, using 4 candle sticks. The accumulated residue, known as bare candle soot, underwent a 15 min sonication process using an ultrasonic bath (Elmasonic P, Elma Schmidbauer GmbH, Singen, Germany) at a frequency of 37 kHz and a power level of 100 W, using a mixture of ethanol (95%, R&M Chemicals, Selangor, Malaysia) and deionized water in a 1:1 proportion. The objective of this washing technique is to effectively remove residual impurities and unburned hydrocarbons from untreated carbon soot [21]. Subsequently, to completely remove the moisture content from the carbon soot, the candle soot that had been washed underwent a drying process at a temperature of 110 °C for the duration of one night. Subsequently, the washed soot underwent functionalization through treatment with nitric acid (HNO_3_, 69%, R&M Chemicals, Selangor, Malaysia) and was subjected to reflux for a duration of 8 h at a temperature of 100 °C. In order to remove any remaining acid, the functionalized soot (CS) was subjected to filtration and subsequent rinsing with deionized water until the effluent achieved a pH level of 7. Subsequently, the sample was subjected to a drying process in an oven set at a temperature of 60 °C for a duration of 12 h, with the aim of removing any remaining moisture.

### 2.2. Synthesis of Ni–CS Electrocatalyst

The Ni–CS electrocatalysts with varying metal weight percentages (5, 10, 15, 20, and 25 wt%) were synthesized using a wet impregnation method [22]. The precursors of nickel were impregnated onto the CS at a temperature of 353 K. Subsequently, the provided sample was subjected to a drying process in an oven at a temperature of 383 K overnight. Following this, the sample was calcined in an inert environment at a temperature of 823 K for a period of three hours. An inert environment is necessary to prevent the oxidation of metals from oxidizing at high temperatures. Using the same steps, a commercial carbon black (CB) (Vulcan XC-72, Cabot Corporation, Massachusetts, MA, USA) supported electrocatalyst was synthesized for comparison.

### 2.3. Structural Characterization of Electrocatalyst

X-ray diffraction (XRD) patterns (Rigaku, Tokyo, Japan) were obtained on a Bruker AXS DB Advance diffractometer (CuKα radiation, λ = 0.154 nm) ranging from 2° to 90°. The surface morphology analysis for the samples was carried out by field emission scanning electron microscopy (FESEM) (JSM-7800F, JEOL, Tokyo, Japan). Scanning electron microscopy–energy dispersive X-ray (SEM–EDX) was tested using JSM-6010plus/LV (JEOL, Tokyo, Japan). Raman spectra (DXR2xi, Thermo Scientific, Waltham, MA, USA) of samples were measured using a laser Raman micro spectrometer using an excitation wavelength of 532 nm.

### 2.4. Working Electrode Preparation

Electrocatalytic inks were prepared by the amalgamation of 10 mg of the catalyst with 150 µL of deionized water, 30 µL of a 5 wt.% Nafion solution (5 wt%, Sigma Aldrich, St. Louis, MO, USA), and 120 µL of 2-propanol (95%, R&M Chemicals, Selangor, Malaysia). The resulting mixture was dispersed for a duration of one hour using an ultrasonic bath. The ink obtained was subsequently transferred in a volume of 2.5 µL onto the surface of a glassy carbon electrode (GCE). Prior to this, the electrode underwent thorough polishing using 0.05 µm alumina pastes, followed by ultrasonic cleaning in deionized water. The electrode was subjected to a drying process overnight prior to conducting measurements.

### 2.5. Electrocatalytic Performance Measurements

The electrocatalysts were subjected to electrochemical analysis using the VersaSTAT 3 (AMETEK Scientific, Westerly, RI, USA) electrochemical workstation. The electrocatalytic activity of the Ni–CS catalysts toward EOR was assessed using cyclic voltammetry (CV) in a standard three-electrode cell. The potential mentioned in this study is presented based on the reversible hydrogen electrode (RHE) scale. The electrochemical characterization is equipped with three types of electrodes (Metrohm, Selangor, Malaysia), namely (i) Working electrode: Glassy carbon electrode (GCE) with a surface area of 0.07068 cm^2^, (ii) Counter electrode: Platinum (Pt) sheet electrode, and (iii) Reference electrode: Ag/AgCl electrode (0.205 V vs. reference hydrogen electrode (RHE)). Prior to conducting CV measurements, a solution consisting of 1.0 M NaOH (Pellets, Sigma-Aldrich, St. Louis, MO, USA) in 1.0 M C_2_H_5_OH was subjected to a 15 min bubbling process with N_2_ saturation to eliminate any presence of oxygen. CV experiments were conducted using a solution comprising an electrolyte of 1.0 M NaOH and 1 M C_2_H_5_OH as fuel. The experiments were carried out at a scan rate of 50 mV/s, with a potential range from −0.2 to 0.6 V vs. Ag/AgCl, all at room temperature. Chronoamperometry (CA) was conducted to measure the stability of the electrocatalyst. CA measurements were conducted at a potential of 0.5 V for a duration of 3600 s.

## 3. Results and Discussion

### 3.1. Structural Characterization

An X-ray diffraction (XRD) analysis was conducted to examine the crystalline structure of the synthesized electrocatalysts, as depicted in Figure 1 and Figure 2. According to Figure 1, the two diffraction peaks at angles of 24.61° and 43.21° are related to the X-ray reflection from the (002) and (100) planes, respectively, which correspond to the graphitic carbon phase [23,24]. The XRD patterns are consistent with the JCPDS card (No-41-1487) of graphite. By comparing Figure 1 and Figure 2A, it can be seen that, after Ni was loaded into candle soot, the peak for (100) planes disappeared. This phenomenon occurred because Ni metal filled up the carbon lattice after metal loading [12]. In Figure 2B, the XRD curve of the Ni–CS catalyst shows several peaks at 2θ of 44.63°, 51.73°, and 76.35°, which correspond to the (111), (200), and (220) planes of cubic Ni crystal (JCPDS No. 47-1049). The intensification of peaks was observed with the augmentation in Ni loading, indicating the presence of a highly crystalline-phase metallic Ni on the CS. However, XRD analysis of the 5 wt% Ni–CS catalyst reveals only one small peak around 43.96°, corresponding to the presence of nickel (Ni) metal. The observed phenomenon can be attributed to the insufficient amount of Ni, which impeded the proper crystallization of metallic Ni. Consequently, the disordered state failed to exhibit any distinct peak in the XRD pattern [25]. Nickel diffraction should be around 44°, due to the broad peak appearing for the metallic nickel in the XRD analysis for 5 wt% at 35–50° which can be attributed to the nanocrystalline nature of the nickel particles. This happens when the crystallite size of a material is small, the XRD peaks become broader due to the limited number of lattice planes that contribute to the diffraction signal, which is known as the Scherrer effect [26,27].

Additionally, the estimation of the average crystallite size of the electrocatalyst is calculated using Scherrer Equation (1), as seen in Figure 2:d _(111)_ = Kλ/β cos θ(1)
where d _(111)_ represents the crystallite size, K symbolizes the Scherrer constant with a value of 0.94, λ denotes the wavelength of the X-ray beam used (1.5418 Å), β is used to describe the full width at half maximum (FWHM), and θ refers to the Bragg angle.

The average size of nickel crystallites exhibits a substantial increase, rising from 13.5 to 25.0 nm, when the Ni loading increases from 10 to 25 wt% (as shown in Table 1). The catalysts with low Ni loading had a narrower size distribution and lower average values compared to the catalysts with high Ni loading. This indicates that there is a greater tendency for Ni particles to agglomerate at higher loading levels [28]. The crystallite size of both 20 wt% and 25 wt% Ni–CS was measured to be 25.0 nm. This is because 20 wt% represents the maximum loading capacity for the candle soot; beyond this point, the Ni NPs can no longer penetrate the carbon pores as they are already saturated with Ni NPs [29]. This means that an excessive quantity of nickel particles present on the catalyst surface will ultimately result in the agglomeration of such particles, which can block the active sites for the electrocatalytic activity.

The Raman spectra of the candle soot, as depicted in Figure 3, reveal the presence of the D and G bands, which are indicative of the relative abundance of sp^3^ carbon (defective sites) and sp^2^ carbon (graphitic) in the sample. The detected peaks at 1350 cm^−1^ and 1590 cm^−1^ correspond to the D band and G band, respectively. The Raman spectra of candle soot show that the D band is weaker and broader than the G band, suggesting a significant number of defects or vacancies caused by the short residence period of soot in flames, which prevents the complete graphitization of soot [30]. The A_D_/A_G_ ratio was computed using the intensity of the G peak which is higher than that of the D peak, and the integration of the area under each peak [31], including its width and shape as 1.62, indicating a higher degree of disorderliness in the candle soot sample [32]. Therefore, an increase in the number of metal-active sites is expected due to the availability of a greater number of surface defects for the attachment of metal particles, as corroborated by the XRD findings. The increase in the number of active sites in electrocatalysts can facilitate the adsorption of reactants and intermediates during EOR, leading to improved catalytic performance. While there is a single hump appearing in the graph, which is attributed to the 2D (G’) band at 2800 cm^−1^, indicating the presence of a few-layer graphene, its displacement is observed to be forward for multilayer graphene [33].

The surface morphological structure of candle soot, 5 wt% Ni–CS, and 20 wt% Ni–CS samples was further determined by field emission scanning electron microscopy (FESEM) (Figure 4). The FESEM image of the candle soot in Figure 4A exhibits the formation of fractal-like spherical carbon nanoparticles that are interconnected to one another [34]. The nanoparticles derived from candle soot exhibit a high degree of branching and partial interconnectivity, with a diameter falling within the range of 14–32 nm, as indicated by the particle size distribution graph. The FESEM images of 5 wt% and 20 wt% Ni–CS are depicted in Figure 4B,C. The size distribution of the latter is comparatively greater than that of the former. The expansion of these values occurs to a certain degree with increasing Ni content, which can be attributed to the comparatively larger atomic radius of Ni in comparison to carbon [35]. Figure 4B shows that there are no visible Ni nanoparticles available in the morphological image. Given the low concentration of nickel in 5 wt% Ni–CS, it is likely that the nickel atoms are impregnated within the carbon matrix (candle soot). At this concentration, the nickel is dispersed at a scale that may not result in distinct crystalline nanoparticles observable by conventional FESEM techniques.

The confirmation of Ni deposition onto candle soot for the 5 wt% Ni–CS sample was established through the utilization of EDX spectra and morphology mapping (Figure 5). It can be seen that Ni successfully loaded into the candle soot after impregnation. The amount of nickel present is minimal, as evidenced by the EDX spectra, which is consistent with the findings of the XRD analysis. The major constituent is carbon, at 98.5%, while nickel accounts for around 1.5%. The distribution of nickel for 5 wt% and 20 wt% Ni–CS catalysts exhibit a high degree of dispersion on the surface of the candle soot through mapping, which can be seen in the supplementary data document (Appendix A).

### 3.2. Electrochemical Characterization

The electrochemical behavior of the prepared electrodes was characterized through the utilization of cyclic voltammetry. The cyclic voltammograms of nickel supported on candle soot with different weight loadings and 20 wt% nickel supported on CB (commercial catalyst) in 1.0 M NaOH solution at a potential range from −0.2 to 0.6 V versus Ag/AgCl at a scan rate of 50 mV/s are shown in Figure 6A. The electrochemical analysis of the Ni–CS electrode revealed the presence of redox peaks at approximately 0.41 V and 0.30 V in the anodic and cathodic directions, respectively. These peaks can be attributed to the Ni^2+^/Ni^3+^ redox couple that was generated on the catalyst surface within the alkaline medium [36]. Based on the electrochemical characteristics of nickel electrodes in alkaline electrolytes, a naturally occurring thin layer of Ni(OH)_2_ is generated on the surface of the nickel. This results in the electrochemical passivation of nickel through the coating of Ni(OH)_2_ [7]. Based on the cyclic voltammetry (CV) curves, the reaction mechanism can be defined as follows [37]:Ni + 2OH^−^ ↔ Ni(OH)_2_ + 2e^−^(2)
Ni(OH)_2_ + OH^−^ → NiOOH + H_2_O + e^−^(3)

It is worth noting that the redox peak currents exhibit a progressive and linear increase when the catalyst loading is increased. This observation confirms the presence of intrinsically active Ni^2+^/Ni^3+^ redox centers [38]. The amount of activity shown by nickel-based electrocatalysts is directly proportional to the quantity of active species generated, which can be determined by the electrochemical surface area (ECSA). The ECSA was determined by employing the corresponding equation [39,40]:ECSA = Q/mq(4)
where Q represents the charge necessary to convert NiOOH to Ni(OH)_2_ during backward scan, m is the catalyst mass, whereas q is equal to 257 μC/cm^2^, which corresponds to the charge associated with the formation of a monolayer of Ni(OH)_2_ from NiOOH. The influence of the nickel content on the ECSA of candle soot is demonstrated in Figure 6B. A higher concentration of nickel in the candle soot leads to a notable improvement in the ECSA. Despite this, it should be noted that the correlation between the ECSA and nickel amount does not exhibit a linear relationship, particularly when the nickel percentage reaches more than 20 wt%. For the 25 wt% sample, the calculated ECSA value was 0.57 m^2^/g, which dropped dramatically compared to the 20 wt% sample of 8.02 m^2^/g. This quantitative aspect is rationalized by the fact that an increase in nickel loading results in a greater number of nickel atoms available to potentially saturate the active site responsible for the reaction, which is in agreement with the results of the crystallite size values from XRD analysis [41]. The formation of Ni(OH)_2_ during the reaction leads to an increase in the concentration of OH^−^ and/or OH_ads_. The excessive concentration of 25 wt% Ni–CS hindered the transfer of ethanol to the active sites on candle soot. If there is an excess of Ni, more Ni will be present on the surface of the candle soot, which may block the active sites of the candle soot and decrease the overall catalytic activity for EOR. In comparison with the commercial catalyst, the ECSA value of 20 wt% Ni–CS is higher than that of 20 wt% Ni–CB (6.80 m^2^/g). This suggests that candle soot as a catalyst support provides more active sites available for the reaction to occur compared to carbon black.

The electrocatalytic performance for the oxidation of ethanol was evaluated in a solution saturated with N_2_, comprising 1 M NaOH and 1 M ethanol, using a scan rate of 50 mV/s. Figure 7A illustrates the CV of different Ni–CS catalysts. In general, ethanol oxidation is indicated by two clearly defined current peaks in the forward and reverse measurements. The oxidation of chemisorbed ethanol species is primarily attributed to the forward current (I_f_). However, during the forward scan, carbonaceous species are not entirely oxidized and are subsequently removed through oxidative means by the reverse scan currents (I_b_) [42]. The forward peak current density of ethanol oxidation exhibits an upward trend with the rise in Ni weight loading, up to a maximum of 20 wt%. An increase in weight loading results in a reduction in electrocatalytic activity. The presence of an excessive quantity of nickel particles appears to result in a decrease in the number of active sites within the material, leading to sluggishness in the electron transfer pathway. It is worth noting that the ideal Ni loading is 20 wt%, which reveals the best electrocatalytic activity of Ni–CS electrocatalysts for ethanol oxidation with a current density of 18.43 mA/cm^2^ in comparison with other synthesized electrocatalysts and 20 wt% Ni–CB. In the presence of higher amounts of nickel, there exists a heightened likelihood of the active sites situated on the surface of the catalyst becoming blocked or covered. This phenomenon may occur due to the adsorption of reaction intermediates or byproducts onto the surface of nickel, which may restrict the access of reactant molecules to the active sites. Thus, it can be inferred that the electrocatalytic activity of ethanol experiences a significant decline at 25 wt% Ni–CS electrocatalyst. Furthermore, it can be noted from Figure 7B that the trend of the maximum current density closely reflects that of the ECSA value, as depicted in Figure 6B. This discovery suggests that the electrooxidation of ethanol on the proposed electrodes is exactly proportional to the value of the ECSA. Increased ECSA generally leads to an increased maximum current density for ethanol electrooxidation. This is because a larger surface area facilitates more active sites for the electrochemical reaction to occur, allowing more ethanol molecules to be oxidized per unit area [43]. The current density performance of EOR would be the same as the mass activity since the trend is similar in both graphs; hence, the result for 20 wt% Ni–CS has the highest mass activity in EOR compared with 20 wt% Ni–CB. This is also supported by the previous study by M. Abdullah et al. [44], who found that the methanol oxidation reaction for Pt-based catalysts had the same trends between current density vs. potential and mass activity vs. potential graphs. For the catalytic activity normalized to the electrode surface area, current density measurements in CV are suitable compared to mass activity.

Table 2 displays the onset potential and current density observed on the Ni–CS electrode during ethanol electrooxidation and comparisons with other studies. The Ni–CS catalyst exhibits promising potential as a feasible choice for methanol electrooxidation when compared to other Ni-based electrodes.

The practical utility of electrodes depends on their stability, in addition to the importance of their high electrocatalytic activity. Chronoamperometry was conducted for 3600 s at 1 M of ethanol and NaOH to assess the stability of all the Ni–CS electrocatalysts, as depicted in Figure 8. The polarization current density of all samples decreased significantly initially and subsequently declined steadily. This is presumably owing to the deactivation of the catalyst surface over time, as the active sites become blocked or poisoned throughout the reaction [46]. The results obtained from chronoamperometry profiles indicate an initial decrease in current density, followed by a quasi-stabilized current density. It was observed that the 20 wt% Ni–CS exhibited the most stability compared to other samples. The decline in current density at the outset can be attributed to the presence of reaction products that are firmly bound to the surface of Ni, thereby hindering the active sites. The stabilization of the active site blockage was ultimately accomplished through achieving the state of equilibrium adsorption/desorption or blocking of a particular preferred site [47]. Moreover, the 20 wt% Ni–CS electrocatalyst recorded the highest retention rate of the other four samples at 65.80%. When compared to the 20wt% Ni at 65.80% CB electrocatalyst, which has a retention rate of 32.20%, the retention rate of the 20wt% Ni–CS is twice as high. This suggests that the 20 wt% Ni–CS electrocatalyst is more efficient in facilitating ethanol oxidation compared to the 20 wt% Ni–CB electrocatalyst. The exceptional performance of the 20 wt% Ni–CS electrocatalyst is attributed to the interconnected fractal-like spherical carbon structure, which facilitates efficient electron transport pathways during ethanol oxidation. The retention rate for different Ni loadings increased from 5 wt% to 20 wt%; however, at 25 wt%, the retention rate dropped. This is probably because excessive Ni concentration blocked the surface candle soot active sites and reduced the passivation of active Ni sites. These findings highlight the potential of 20 wt% Ni–CS as a superior choice for ethanol oxidation applications.

## 4. Conclusions

In summary, we successfully fabricated carbon nanoparticles by turning soot pollutants from candles through a simple and cost-friendly approach, which were subsequently decorated with Ni nanoparticles. The findings of the comparative analysis conducted on five distinct electrocatalysts revealed that 20 wt% Ni–CS exhibited the most superior efficiency in facilitating ethanol oxidation, outperforming commercial 20 wt% Ni–CB. The enhanced performance of Ni–CS can be attributed primarily to its mesoporous structure and the high degree of nickel dispersion within the material. The stability of 20 wt% Ni–CS with a retention rate of 65.80% was demonstrated through chronoamperometry estimations. The electrode developed by combining the superior electrochemical characteristics of candle soot and nickel nanoparticles exhibits promising potential for ethanol oxidation, owing to its remarkable stability and performance. 

## Figures and Tables

**Figure 1 nanomaterials-14-01042-f001:**
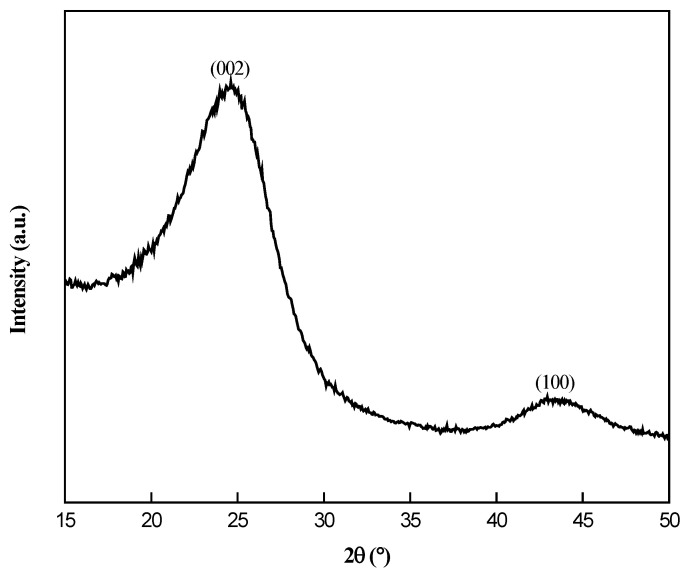
XRD patterns of CS.

**Figure 2 nanomaterials-14-01042-f002:**
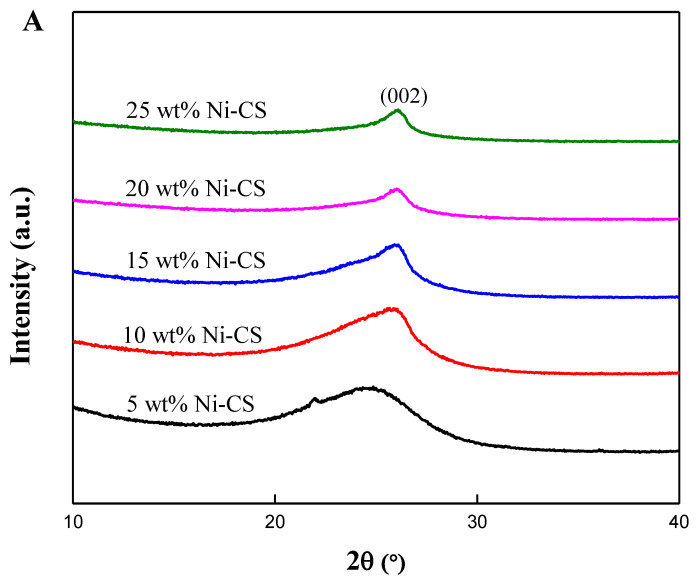
(**A**) Magnification of (**B**) XRD patterns of 5 wt%, 10 wt%, 15 wt%, 20 wt%, and 25 wt% Ni–CS catalysts.

**Figure 3 nanomaterials-14-01042-f003:**
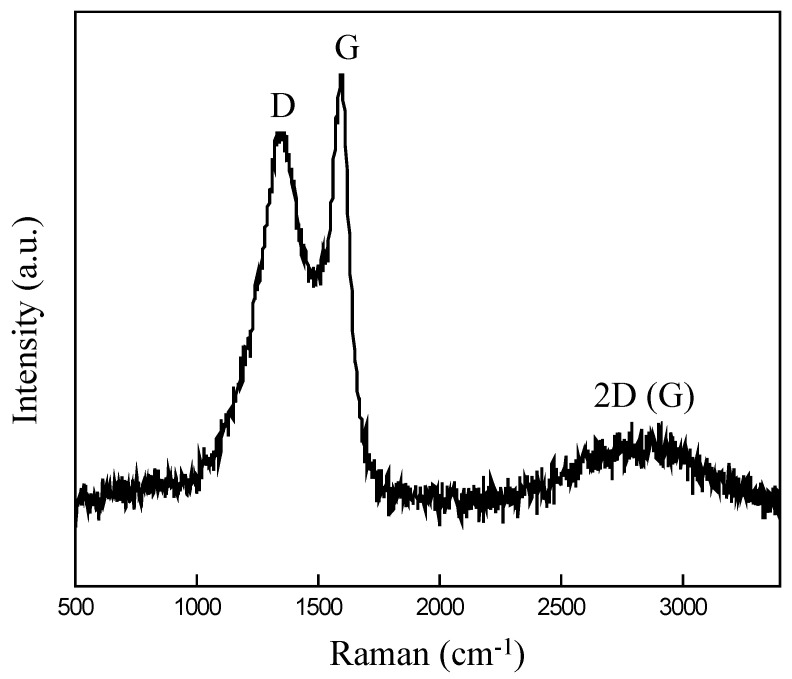
Raman spectra of the CS.

**Figure 4 nanomaterials-14-01042-f004:**
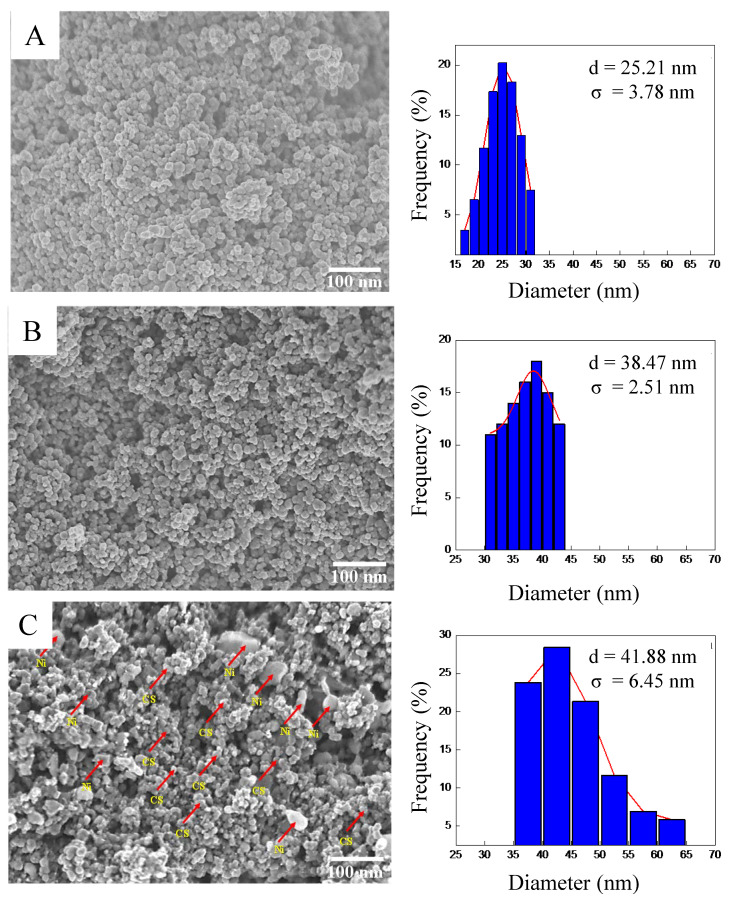
FESEM images and particle size distribution of (**A**) candle soot, (**B**) 5 wt%, and (**C**) 20 wt% Ni–CS catalysts.

**Figure 5 nanomaterials-14-01042-f005:**
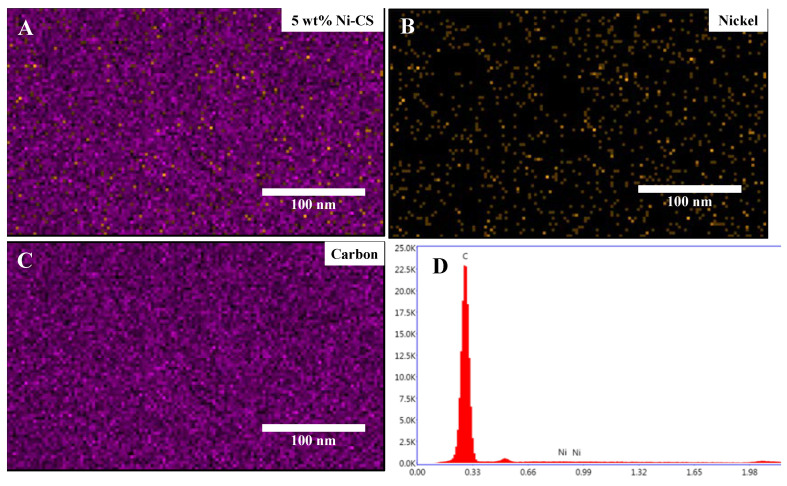
(**A**) Mapping images, (**B**) Nickel element, (**C**) Carbon element, and (**D**) EDX of 5 wt% Ni–CS catalysts.

**Figure 6 nanomaterials-14-01042-f006:**
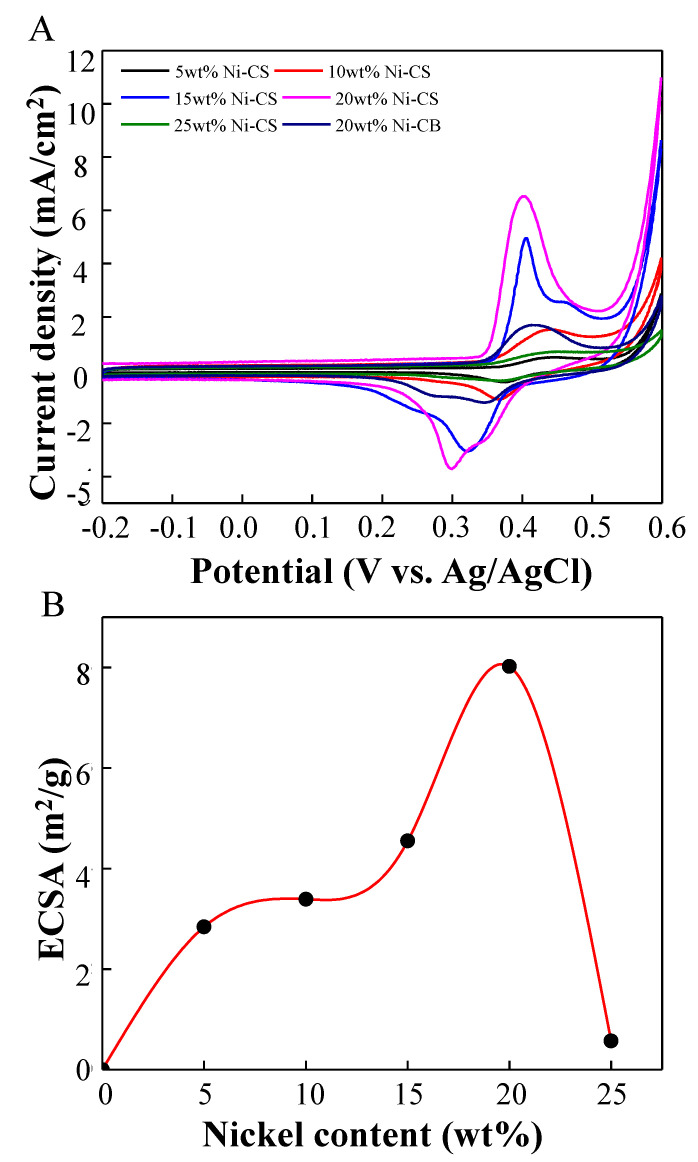
Cyclic voltammograms of 5 wt%, 10 wt%, 15 wt%, 20 wt%, and 25 wt% Ni–CS and 20 wt% Ni–CB catalysts in (**A**) 1.0 M NaOH at scan rate of 50 mVs^−1^ and (**B**) effect of the nickel content on the ECSA value.

**Figure 7 nanomaterials-14-01042-f007:**
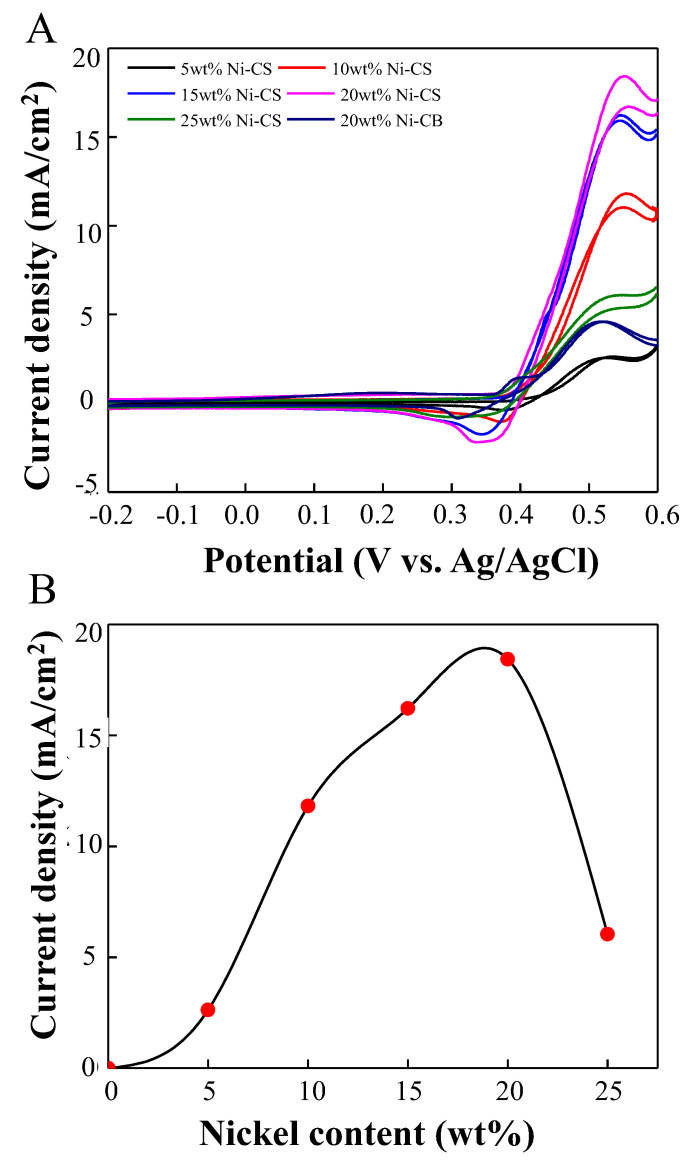
Cyclic voltammograms of 5 wt%, 10 wt%, 15 wt%, 20 wt%, and 25 wt% Ni–CS, and 20 wt% Ni–CB catalysts in (**A**) 1.0 M NaOH + 1.0 M CH_3_CH_2_OH at a scan rate of 50 mV/s and (**B**) effect of the nickel content on the current density.

**Figure 8 nanomaterials-14-01042-f008:**
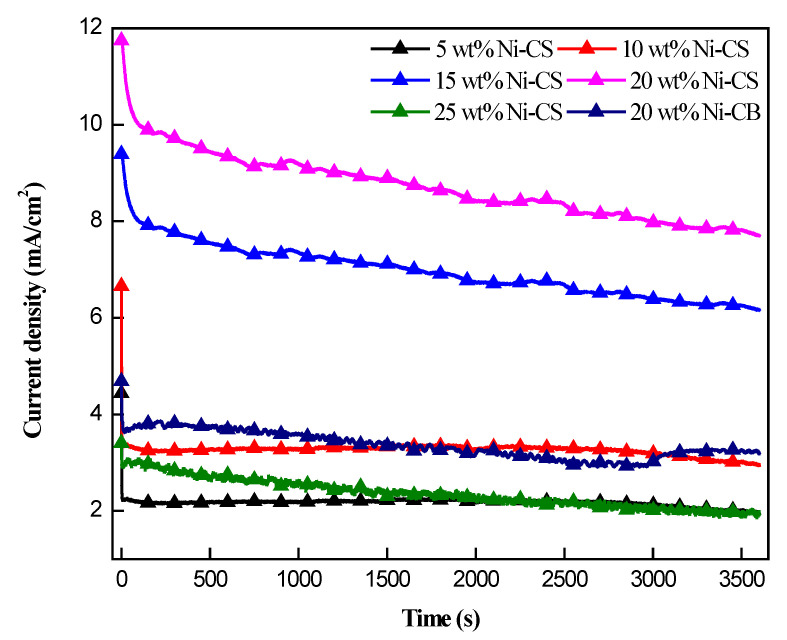
Chronoamperometry of 5 wt%, 10 wt%, 15 wt%, 20 wt%, and 25 wt% Ni–CS, and 20wt% Ni–CB catalysts in 1.0 M NaOH + 1.0 M CH_3_CH_2_OH at potential of 0.5 V.

**Table 1 nanomaterials-14-01042-t001:** Values of 2θ, FWHM, and Ni^0^ crystallite sizes for all the electrocatalysts determined from XRD studies.

Electrocatalyst	2θ (111) (°)	FWHM (°)	Ni^0^ Crystallite Sizes (nm)
5 wt% Ni–CS	-	-	-
10 wt% Ni–CS	44.59	0.6652	13.5
15 wt% Ni–CS	44.50	0.4606	19.5
20 wt% Ni–CS	44.50	0.3582	25.0
25 wt% Ni–CS	44.49	0.3582	25.0

**Table 2 nanomaterials-14-01042-t002:** Comparisons of electrocatalytic performance with the previous study.

Electrocatalyst	Conditions	ECSA (m^2^/g)	Onset Potential (V)	I_F_ (mA/cm^2^)	Ref.
Ni/Al_2_O_3_–5/GC	0.1 M NaOH + 0.1 M CH_3_OH	NA	0.72 vs. Ag/AgCl	11.1	[7]
MOF1-CDs@CC	1 M KOH + 1 M CH_3_CH_2_OH	NA	1.3 vs. RHE	119	[45]
Ni/NiO/MWCNT	0.1 M NaOH + 0.7 M CH_3_OH	4.52	0.43 vs. Ag/AgCl	15.94	[8]
NiNPs-R/PPy/CPE	0.1 M NaOH + 0.2 M CH_3_CH_2_OH	NA	0.80 vs. Ag/AgCl	21.1	[9]
20 wt% Ni-CB	1 M NaOH + 1 M CH_3_CH_2_OH	6.80	0.36 vs. Ag/AgCl	4.60	This study
20 wt% Ni-CS	1 M NaOH + 1 M CH_3_CH_2_OH	8.02	0.37 vs. Ag/AgCl	18.43	This study

Note: NA—not available.

## Data Availability

Data are contained within the article and Appendix A.

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
