# Peer review of "Candle Soot as a Novel Support for Nickel Nanoparticles in the Electrocatalytic Ethanol Oxidation"

_nanomaterials, 2024, doi:10.3390/nano14121042_

Round 1

Reviewer 1 Report (Previous Reviewer 1)

Comments and Suggestions for Authors

This revised manuscript addressed the comments from the previous review in a reasonable way. However, more explanations are expected to illustrate the mismatch in the characterization for 5 wt% Ni-CS in terms of XRD, SEM and EDS mapping.

1. It’s suggested to point out the NiNPs in Figure 4 (as in the original version) to help the audience identify the NiNPs on the CS support.

2. With the scale bar added in EDS mapping images in Figure 5, the Ni content in 5 wt.% Ni-CS catalysts is more like single atoms instead of NiNPs with average size if 38.47nm, which should be obvious in the mapping image.

3. Despite the effort the discover the peak at 35-50 ° (not °C as in the Response to Review 1 Comments), it’s too wide to be identified as the Ni  diffraction peak.

4. Please add EDS mapping for 20 wt% Ni-CS catalysts. SEM images (Figure 4C) and EDS mapping images (to be added) should be the same area so that the NiNPs and CS can easily be distinguished. 

Author Response

Dear reviewer 1,

Reviewer 2 Report (New Reviewer)

Comments and Suggestions for Authors

The proposed candle soot as a carbon support may not good for industrial scale, it's not a reasonable material resource. The authors may need to check the ORR (oxygen reduction reaction) for the proposed electrochemical electrode composed from the carbon support. The ECSA seems not a good data compared to other carbon supports, the authors may check them.   

Author Response

Dear reviewer 2,

Reviewer 3 Report (New Reviewer)

Comments and Suggestions for Authors

The authors presented Ni as a catalyst for ethanol oxidation and wanted to show the potential application of a new carbon carrier, candle soot carbon.

The experiments, conducted in line with the authors' objectives, are a significant step in understanding electrochemical ethanol oxidation reactions by immersing catalysts with different amounts of Ni in CS.

While the current organization of the paper may need some adjustments, I am confident that with the following comments, the paper will be ready for publication.

 1.     The authors propose a new CS as a carbon carrier, but only XRD, Raman, and SEM are provided for physical data on the properties of the CS. It would be nice to provide data on nitrogen adsorption experiments to understand the specific surface area and pore distribution, which are important as a support.

2.     I think it would be good to increase the number of references to CS in the introduction (Lines 103-105 in the manuscript)

3.     The authors have described the process of obtaining CS in the experimental part, please provide the yield of this carbon per hour. (Lines 132-133)

4.     In the experimental part, a cellulose sample is mentioned, which I think needs further explanation. (Line 144)

5.     The authors should explain why they used the oxygen environments for the catalyst preparation. (Lines 152-154)

6.     It is better to merge the Figure 1 to the Figure 2a. (lines 192-194).

7.     The intensity of the G peak is higher than that of the D peak, which induces the ID/IG to be less than 1. However, the authors provide a value of 1.62. They need to elucidate this value. (Lines 234-237)

8.     This is not an acceptable explanation for the increase in the average particle sizes of candle shoot carbon. (Lines 253-255) In addition, the authors should confirm the values for the crystalline size of the 25 wt% Ni-CS. It is the same as the 20 wt% Ni-CS. The average particle sizes of the catalyst should be determined using transmission electron microscopy.

9.     This is not as important to show as in Figure 5. It should be rearranged to the supplementary data if there are TEM images. (Line 271)

10.  In Figure 6A, Voltage should be converted to value vs. RHE. (Line 315) Also, I would like the author to increase the upper limit of voltage to show the whole oxidation of the ethanol.

11.  The authors should consider the mass activity or specific activity of the Ni/CS and Ni/CB catalysts for ethanol electrooxidation.

12. It is better to confirm the font, such as superscript and subscript, for the chemicals in the titles of references. 

Comments on the Quality of English Language

It need to be refined.

Author Response

Dear reviewer 3,

Round 2

Reviewer 1 Report (Previous Reviewer 1)

Comments and Suggestions for Authors

The authors have addressed most of the comments. It's suggested to consider acceptance of publication on Nanomaterials as in the current form.

Author Response

Dear Reviewer,

Please find the reviewer report in the attachment.

Thank you very much for your time.

Reviewer 3 Report (New Reviewer)

Comments and Suggestions for Authors

The authors have effectively addressed most of the reviewers' comments. However, two aspects require further revision.

Firstly, the reported specific surface area of the carbon black appears smaller than the typical range, necessitating further verification. If a commercial carbon black was used, detailed product information would be beneficial. Additionally, providing the specific surface area of the carbon black (CB) used for Pt/CB, as mentioned in the paper, is recommended.

Secondly, the authors stated that ID/IG was obtained by comparing the areas of the peaks. In this case, it's suggested to use A (for area) instead of I for notation and express it as AD/AG. This change would not only enhance clarity but also align with the standard notation used in the field.

Comments on the Quality of English Language

Careful proofreading is required.

Author Response

Dear Reviewer,

Please find the reviewer report in the attachment.

Thank you very much for your time.

This manuscript is a resubmission of an earlier submission. The following is a list of the peer review reports and author responses from that submission.

Round 1

Reviewer 1 Report

Comments and Suggestions for Authors

This paper utilized candle soot (CS) as a novel and accessible carbon support for Ni nanoparticles (NiNPs) and employed wet impregnation method for synthesis of the catalysts for ethanol oxidation. Despite the seemingly thorough characterization on the morphologies and electrochemical performance of the catalysts. A deeper insight into the reaction mechanism regarding the reported high activity and stability of the 20 wt.% Ni-CS is desired as well as potentially more exploration on the structural advantages of CS over carbon black and other forms carbon, since this is claimed as one of the novelty and originality of this paper. Below comments need to be addressed before considering acceptance of this paper for publication on Nanomaterials.

1.    A scale bar is needed for EDX mapping images in Fig. 5, in which the Ni signals are weak and almost single-unit pixels, which is contradictory to the noticeable Ni particles and large size distribution (30-44nm) as in Fig. 4 (line 257).

2.    The size distribution range is much larger on the 20 wt.% Ni-CS than the 5 wt.% Ni-CS, while the average diameter is only ~1.5nm larger. In the FESEM image of 20wt% Ni-CS, certain NiNPs are much larger than 50nm but not revealed in size distribution histogram. The data is presented is not accurate.

3.   For 5 wt.% Ni-CS catalyst with average NiNPs diameter of 38.49nm, the crystallization should be significant in XRD pattern. However, it’s surprising that no Ni peaks were detected as in Fig. 2B.

4.    The authors claimed the advantage of CS as carbon support is due to it’s high disorderliness? What kind of bond is formed between Ni and CS? How does the defective sites affect Ni-CS electrocatalytic performance?

5.  In line 296-301, the authors stated that ‘an increase in nickel loading results in a greater number of nickel atoms available to potentially saturate the active site responsible for the reaction, which is in agreement with the results of the crystallite size value from XRD analysis’. However, the XRD results in Figure 2B and Table 1 indicated that 20 wt% Ni-CS and 25 wt% Ni-CS have the same Ni crystalline size. It’s not a solid statement for explaining the dramatically dropped ECSA of 0.57 m2/g in 25 wt% Ni-CS compared to 8.02 m2/g in 20 wt.% Ni-CS.

6.     A sharp drop in CA was noticed for 5 wt.%, 10 wt%, 25 wt.% Ni-CS and 20 wt% Ni-CB at t=0s in Fig. 8. What’s the reason for the sharp drop. Does this indicate loss of active sites on the NiNPs or dissolution of NiNPs on CS? Please add more supportive info to this sharp drop.

7.     What’s the potential of Ag/AgCl vs. RHE in this paper? This info should be provided and is important for peers to repeat the experiments and cross-check electrochemical performances as presented in the paper.

Comments on the Quality of English Language

Some typo and grammar errors exist in the manuscript. 

Reviewer 2 Report

Comments and Suggestions for Authors

1. The authors elaborate on the relationship between maximum current density and electrochemical surface area (ECSA) and how this relationship affects the understanding of ethanol electrooxidation at these electrodes?

2. Can the authors elaborate on the observed differences in retention between the Ni-CS and Ni-CB electrocatalysts, and in particular the superior stability and efficiency of the 20 wt% Ni-CS electrocatalyst for ethanol oxidation?

3. The following literature is suggested to consolidate the presentation.

4. Journal of Power Sources. 2024, 590, 233801, Journal of the Taiwan Institute of Chemical Engineers, 2014, 45(4), 1532-1541, International Journal of Hydrogen Energy. 2024, 52, 1105-1114。

5. What are the main conclusions drawn from the time-varying analysis of the stability of nickel-loaded candle smoke (Ni-CS) electrocatalysts for ethanol oxidation, as shown in Fig. 8?

6.  Specifically, how does the weight loading of nickel affect the stability of the electrocatalysts and what are the implications of these findings for practical applications?

7.  Do the authors plan to conduct further research?

Comments on the Quality of English Language

can be improved